# Unsupervised Dense Information Retrieval with Contrastive Learning

**Gautier Izacard**$^{\diamond,\clubsuit,\heartsuit}$                                            *gizacard@fb.com*
**Mathilde Caron**$^{\diamond,\heartsuit,\spadesuit}$                                            *mathilde@fb.com*
**Lucas Hosseini**$^{\diamond}$                                            *hoss@fb.com*
**Sebastian Riedel**$^{\diamond,\triangle}$                                            *sriedel@fb.com*
**Piotr Bojanowski**$^{\diamond}$                                            *bojanowski@fb.com*
**Armand Joulin**$^{\diamond}$                                            *ajoulin@fb.com*
**Edouard Grave**$^{\diamond}$                                            *egrave@fb.com*
$^{\diamond}$ *Meta AI Research,* $^{\clubsuit}$ *Ecole normale supérieure, PSL University,* $^{\heartsuit}$ *Inria,*
$^{\spadesuit}$ *Université Grenoble Alpes,* $^{\triangle}$ *University College London*

**Reviewed on OpenReview:** *https://openreview.net/forum?id=jKN1pXi7b0*

## Abstract

Recently, information retrieval has seen the emergence of dense retrievers, using neural networks, as an alternative to classical sparse methods based on term-frequency. These models have obtained state-of-the-art results on datasets and tasks where large training sets are available. However, they do not transfer well to new applications with no training data, and are outperformed by unsupervised term-frequency methods such as BM25. In this work, we explore the limits of contrastive learning as a way to train unsupervised dense retrievers and show that it leads to strong performance in various retrieval settings. On the BEIR benchmark our unsupervised model outperforms BM25 on 11 out of 15 datasets for the Recall@100. When used as pre-training before fine-tuning, either on a few thousands in-domain examples or on the large MS MARCO dataset, our contrastive model leads to improvements on the BEIR benchmark. Finally, we evaluate our approach for multi-lingual retrieval, where training data is even scarcer than for English, and show that our approach leads to strong unsupervised performance. Our model also exhibits strong cross-lingual transfer when fine-tuned on supervised English data only and evaluated on low resources language such as Swahili. We show that our unsupervised models can perform cross-lingual retrieval between different scripts, such as retrieving English documents from Arabic queries, which would not be possible with term matching methods.

## 1 Introduction

Document retrieval is the task of finding relevant documents in a large collection to answer specific queries. This is an important task by itself and a core component to solve many natural language processing (NLP) problems, such as open domain question answering (Chen et al., 2017a) or fact checking (Thorne et al., 2018). Traditionally, retrieval systems, or retrievers, leverage lexical similarities to match queries and documents, using, for instance, TF-IDF or BM25 weighting (Robertson & Zaragoza, 2009). These approaches, based on near-exact matches between tokens of the queries and documents, suffer from the lexical gap and do not generalize well (Berger et al., 2000). By contrast, approaches based on neural networks allow learning beyond lexical similarities, resulting in state-of-the-art performance on question answering benchmarks, such as MS MARCO (Nguyen et al., 2016) or NaturalQuestions (Kwiatkowski et al., 2019).

The strong retrieval results of neural networks have been possible for domains and applications where large training datasets are available. In the case of retrieval, creating these datasets requires manually matching queries to the relevant documents in the collection. This is hardly possible when the collection

contains millions or billions of elements, resulting in many scenarios where only a few in-domain examples, if any, are available. A potential solution is to train a dense retriever on a large retrieval dataset such as MS MARCO, and then apply it to new domains, a setting referred to as *zero-shot*. Unfortunately, in this setting, dense retrievers are often outperformed by classical methods based on term-frequency, which do not require supervision (Thakur et al., 2021). Moreover large annotated datasets are generally not available in languages other than English. Thus, using large collections of supervised data is not suitable to train multilingual retrieval systems.

A natural alternative to transfer learning is unsupervised learning, which raises the following question: *is it possible to train dense retrievers without supervision, and match the performance of BM25?* Training dense retrievers without supervision can be achieved by using an auxiliary task that approximates retrieval. Given a document, one can generate a synthetic query and then train the network to retrieve the original document, among many others, given the query. The inverse Cloze task (ICT), proposed by Lee et al. (2019) to pre-train retrievers, uses a given sentence as a query and predicts the context surrounding it. While showing promising results as pre-training (Chang et al., 2020; Sachan et al., 2021), this approach still lags behind BM25 when used as a zero-shot retriever. ICT is strongly related to contrastive learning (Wu et al., 2018), which has been widely applied in computer vision (Chen et al., 2020; He et al., 2020). In particular, computer vision models trained with the latest contrastive learning methods led to features well suited to retrieval (Caron et al., 2021). We thus propose to revisit how well contrastive learning performs to train dense retrievers without supervision.

In this paper, we make the following contributions. First, we show that contrastive learning can lead to strong unsupervised retrievers: our model achieves Recall@100 results competitive with BM25 on most of the BEIR benchmark. Second, in a few-shot setting, we show that our model benefits from few training examples, and obtains better results than transferring models from large datasets such as MS MARCO. Third, when used as a pre-training method before fine-tuning on MS MARCO, our technique leads to strong performance on the BEIR benchmark. We perform ablations to motivate our design choices, and show that cropping works better than the inverse Cloze task. Finally we train a multilingual dense retriever with contrastive learning and show that it achieves state-of-the-art performance.

Code and pre-trained models are available here: `https://github.com/facebookresearch/contriever`.

## 2 Related work

In this section, we briefly review relevant work in information retrieval, and application of machine learning to this problem. This is not an exhaustive review, and we refer the reader to Manning et al. (2008), Mitra et al. (2018) and Lin et al. (2020) for a more complete introduction to the field.

**Term-frequency based information retrieval.** Historically, in information retrieval, documents and queries are represented as sparse vectors where each element of the vectors corresponds to a term of the vocabulary. Different weighing schemes have been proposed, to determine how important a particular term is to a document in a large dataset. One of the most used weighing scheme is known as TF-IDF, and is based on inverse document frequency, or term specificity (Jones, 1972). BM25, which is still widely used today, extends TF-IDF (Robertson et al., 1995). A well known limitation of these approaches is that they rely on near-exact match to retrieve documents. This led to the introduction of latent semantic analysis (Deerwester et al., 1990), in which documents are represented as low dimensional dense vectors.

**Neural network based information retrieval.** Following the successful application of deep learning methods to natural language processing, neural networks techniques were introduced for information retrieval. Huang et al. (2013) proposed a deep bag-of-words model, in which representations of queries and documents are computed independently. A relevance score is then obtained by taking the dot product between representations, and the model is trained end-to-end on click data from a search engine. This method was later refined by replacing the bag-of-words model by convolutional neural networks (Shen et al., 2014) or recurrent neural network (Palangi et al., 2016). A limitation of **bi-encoders** is that queries and documents are represented by a single vector, preventing the model to capture fine-grained interactions between terms. Nogueira & Cho

(2019) introduced a **cross-encoder** model, based on the BERT model (Devlin et al., 2019), which jointly encodes queries and documents. The application of a strong pre-trained model, as well as the cross-encoder architecture, lead to important improvement on the MS MARCO benchmark (Bajaj et al., 2016).

The methods described in the previous paragraph were applied to re-rank documents, which were retrieved with a traditional IR system such as BM25. Gillick et al. (2018) first studied whether continuous retrievers, based on bi-encoder neural models, could be viable alternative to re-ranking. In the context of question answering, Karpukhin et al. (2020) introduced a dense passage retriever (DPR) based on the bi-encoder architecture. This model is initialized with a BERT network, and trained discriminatively using pairs of queries and relevant documents, with hard negatives from BM25. Xiong et al. (2020) further extended this work by mining hard negatives with the model itself during optimization, and trained on the MS MARCO dataset. Once a collection of documents, such as Wikipedia articles, is encoded, retrieval is performed with a fast k-nearest neighbors library such as FAISS Johnson et al. (2019). To alleviate the limitations of bi-encoders, Humeau et al. (2019) introduces the poly-encoder architecture, where documents are encoded by multiple vectors. Similarly, Khattab et al. (2020) proposes the ColBERT model, which keeps a vector representation for each term of the queries and documents. To make the retrieval tractable, the term-level function is approximated to first retrieve an initial set of candidates, which are then re-ranked with the true score. In the context of question answering, knowledge distillation has been used to train retrievers, either using the attention scores of the reader of the downstream task as synthetic labels (Izacard & Grave, 2020a), or the relevance score from a cross encoder (Yang & Seo, 2020). Luan et al. (2020) compares, theoretically and empirically, the performance of sparse and dense retrievers, including bi-, cross- and poly-encoders. Dense retrievers, such as DPR, can lead to indices weighing nearly 100GB when encoding document collections such as Wikipedia. Izacard et al. (2020) shows how to compress such indices, with limited impact on performance, making them more practical to use.

**Self-supervised learning for NLP.** Following the success of word2vec (Mikolov et al., 2013), many self-supervised techniques have been proposed to learn representation of text. Here, we briefly review the ones that are most related to our approach: sentence level models and contrastive techniques. Jernite et al. (2017) introduced different objective functions to learn sentence representations, including next sentence prediction and sentence order prediction. These objectives were later used in pre-trained models based on transformers, such as BERT (Devlin et al., 2019) and AlBERT (Lan et al., 2019). In the context of retrieval, Lee et al. (2019) introduced the inverse cloze task (ICT), whose purpose is to predict the context surrounding a span of text. Guu et al. (2020) integrated a bi-encoder retriever model in a BERT pre-training scheme. The retrieved documents are used as additional context in the BERT task, and the whole system is trained end-to-end in an unsupervised way. Similarly, Lewis et al. (2020) proposed to jointly learn a retriever and a generative seq2seq model, using self-supervised training. Chang et al. (2020) compares different pre-training tasks for retrieval, including the inverse cloze task. In the context of natural language processing, Fang et al. (2020) proposed to apply MoCo where positive pairs of sentences are obtained using back-translation. Different works augmented the masked language modeling objective with a contrastive loss (Giorgi et al., 2020; Wu et al., 2020; Meng et al., 2021). SBERT (Reimers & Gurevych, 2019) uses a Siamese network similar to contrastive learning to learn a BERT-like model that is adapted to matching sentence embeddings. Their formulation is similar to our work but requires aligned pairs of sentences to form positive pairs while we propose to use data augmentation to leverage large unaligned textual corpora. Concurrent to this work, Gao & Callan (2021) have also shown the potential of contrastive learning for information retrieval; building on the same observation that both tasks share a similar structure. Spider (Ram et al., 2021), a contemporary work, uses spans appearing multiple times in a document to create pseudo examples for contrastive learning in order to train unsupervised retrievers. Finally, Chen et al. (2021) train a dense retriever to imitate unsupervised lexical-based methods. This improves performance on a range of tasks and achieves state-of-the-art results when combining the resulting dense retriever with Contriever, our model pre-trained with contrastive learning.

## 3 Method

In this section, we describe how to train a dense retriever with no supervision. We review the model architecture and then describe contrastive learning — a key component of its training.

The objective of a retriever is to find relevant documents in a large collection for a given query. Thus, the retriever takes as input the set of documents and the query and outputs a relevance score for each document. A standard approach is to encode each query-document pair with a neural network (Nogueira & Cho, 2019). This procedure requires re-encoding every document for any new query and hence does not scale to large collections of documents. Instead, we follow standard approaches (Huang et al., 2013; Karpukhin et al., 2020) in information retrieval and use a bi-encoder architecture where documents and queries are encoded independently. The relevance score between a query and a document is given by the dot product between their representations after applying the encoder. More precisely, given a query $q$ and document $d$, we encode each of them independently using the same model, $f_\theta$, parameterized by $\theta$. The relevance score $s(q, d)$ between a query $q$ and a document $d$ is then the dot product of the resulting representations:

$$s(q, d) = \langle f_\theta(q), f_\theta(d) \rangle.$$

In practice, we use a transformer network for $f_\theta$ to embed both queries and documents. Alternatively, two different encoders can be used to encode queries and documents respectively as in DPR (Karpukhin et al., 2020). Empirically, we observed that using the same encoder, such as in Xiong et al. (2020) and Reimers & Gurevych (2019), generally improves robustness in the context of zero-shot transfer or few-shot learning, while having no impact on other settings. Finally, the representation $f_\theta(q)$ (resp. $f_\theta(d)$) for a query (resp. document) is obtained by averaging the hidden representations of the last layer. Following previous work on dense retrieval with neural networks, we use the BERT base uncased architecture and refer the reader to Devlin et al. (2019) for more details.

### 3.1 Unsupervised training on unaligned documents

In this section, we describe our unsupervised training method. We briefly review the loss function traditionally used in contrastive learning and also used in ICT (Lee et al., 2019). We then discuss obtaining positive pairs from a single text document, a critical ingredient for this training paradigm.

#### 3.1.1 Contrastive learning

Contrastive learning is an approach that relies on the fact that every document is, in some way, unique. This signal is the only information available in the absence of manual supervision. A contrastive loss is used to learn by discriminating between documents. This loss compares either positive (from the same document) or negative (from different documents) pairs of document representations. Formally, given a query $q$ with an associated positive document $k_+$, and a pool of negative documents $(k_i)_{i=1..K}$, the contrastive InfoNCE loss is defined as:

$$\mathcal{L}(q, k_+) = -\frac{\exp(s(q, k_+)/\tau)}{\exp(s(q, k_+)/\tau) + \sum_{i=1}^{K} \exp(s(q, k_i)/\tau)}, \tag{1}$$

where $\tau$ is a temperature parameter. This loss encourages positive pairs to have high scores and negative pairs to have low scores. Another interpretation of this loss function is the following: given the query representation $q$, the goal is to recover, or retrieve, the representation $k_+$ corresponding to the positive document, among all the negatives $k_i$. In the following, we refer to the left-hand side representations in the score $s$ as queries and the right-hand side representations as keys.

#### 3.1.2 Building positive pairs from a single document

A crucial element of contrastive learning is how to build positive pairs from a single input. In computer vision, this step relies on applying two independent data augmentations to the same image, resulting in two "views" that form a positive pair (Wu et al., 2018; Chen et al., 2020). While we consider similar independent text transformation, we also explore dependent transformations designed to reduce the correlation between views.

**Inverse Cloze Task** is a data augmentation that generates two mutually exclusive views of a document, introduced in the context of retrieval by Lee et al. (2019). The first view is obtained by randomly sampling a span of tokens from a segment of text, while the complement of the span forms the second view. Specifically, given a sequence of text $(w_1, ..., w_n)$, ICT samples a span $(w_a, ..., w_b)$, where $1 \leq a \leq b \leq n$, uses the

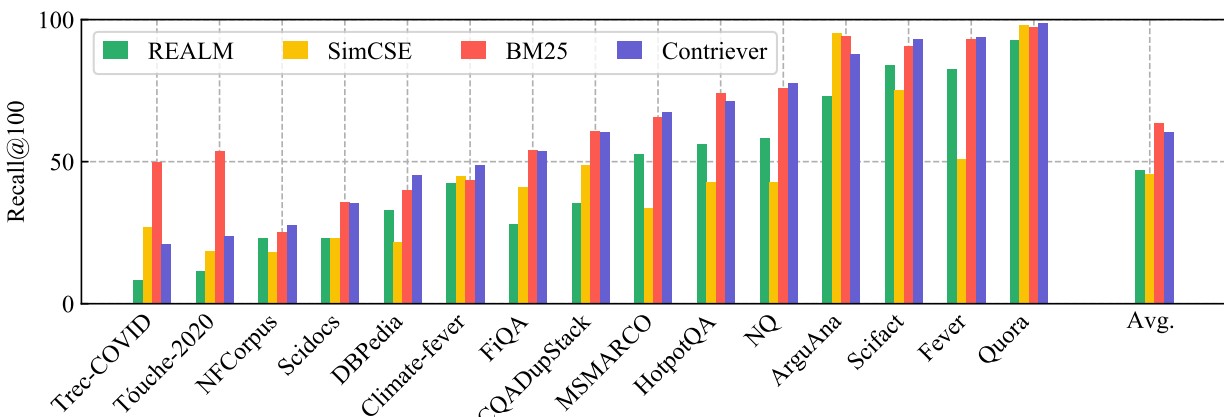

Figure 1: **Unsupervised retrieval.** We compare our pre-training without using *any* annotated data to REALM (Guu et al., 2020), SimCSE (Gao et al., 2021) and BM25. For SimCSE we report results of the model using RoBERTa large. REALM uses annotated entity recognition data for training. We highlight that our unsupervised pre-training is on par with BM25 but on 2 datasets.

tokens of the span as the query and the complement $(w_1, ..., w_{a-1}, w_{b+1}, ..., w_n)$ as the key. In the original implementation by Lee et al. (2019) the span corresponds to a sentence, and is kept in the document 10% of the time to encourage lexical matching. The Inverse Cloze Task is closely related to the Cloze task which uses the span complement $(w_1, ..., w_{a-1}, w_{b+1}, ..., w_n)$ as the query.

**Independent cropping** is a common independent data augmentation used for images where views are generated independently by cropping the input. In the context of text, cropping is equivalent to sampling a span of tokens. This strategy thus samples independently two spans from a document to form a positive pair. As opposed to the inverse Cloze task, in *cropping* both views of the example correspond to contiguous subsequence of the original data. A second difference between cropping and ICT is the fact that independent random cropping is symmetric: both the queries and documents follow the same distribution. Independent cropping also lead to overlap between the two views of the data, hence encouraging the network to learn exact matches between the query and document, in a way that is similar to lexical matching methods like BM25. In practice, we can either fix the length of the span for the query and the key, or sample them.

**Additional data augmentation.** Finally, we also consider additional data augmentations such as random word deletion, replacement or masking. We use these perturbations in addition to random cropping.

### 3.1.3 Building large set of negative pairs

An important aspect of contrastive learning is to sample a large set of negatives. Most standard frameworks differ from each other in terms of how the negatives are handled, and we briefly describe two of them, in-batch negative sampling and MoCo, that we use in this work.

**Negatives within a batch.** A first solution is to generate the negatives by using the other examples from the same batch: each example in a batch is transformed twice to generate positive pairs, and we generate negatives by using the views from the other examples in the batch. We will refer to this technique as "in-batch negatives". In that case, the gradient is back-propagated through the representations of both the queries and the keys. A downside of this approach is that it requires extremely large batch sizes to work well Chen et al. (2020), with Qu et al. (2021) reporting improvement in the context of information retrieval up to 8192 negatives. This method has been widely used to train information retrieval models with supervised data Chen et al. (2017b); Karpukhin et al. (2020) and was also considered when using ICT to pre-train retrievers by Lee et al. (2019).

**Negative pairs across batches.** An alternative approach is to store representations from previous batches in a queue and use them as negative examples in the loss (Wu et al., 2018). This allows for smaller batch size but slightly changes the loss by making it asymmetric between "queries" (one of the view generated from the elements of the current batch), and "keys" (the elements stored in the queue). Gradient is only backpropagated through the "queries", and the representation of the "keys" are considered as fixed. In practice, the features stored in the queue from previous batches comes form previous iterations of the network. This leads to a drop of performance when the network rapidly changes during training. Instead, He et al. (2020) proposed to generate representations of keys from a second network that is updated more slowly. This approach, called MoCo, considers two networks: one for the keys, parametrized by $\theta_k$, and one of the query, parametrized by $\theta_q$. The parameters of the query network are updated with backpropagation and stochastic gradient descent, similarly to when using in-batch negatives, while the parameters of the key network, or Momentum encoder, is updated from the parameters of the query network by using a exponential moving average:

$$\theta_k \leftarrow m\theta_k + (1-m)\theta_q, \tag{2}$$

where $m$ is the momentum parameter that takes its value in $[0, 1]$.

## 4 Experiments

In this section, we empirically evaluate our best retriever trained with contrastive learning, called *Contriever* (contrastive retriever), which uses MoCo with random cropping. We use a contrastive learning procedure that differs from ICT (Lee et al., 2019) mainly in three aspects. First, positive pairs are sampled using random cropping and tokens from each element of the pair are deleted with a probability of 10%. Second we use MoCo where negatives consists of elements from previous batches stored in a queue. This allows to scale to a large number of negatives. Third we use data from Wikipedia and CCNet (Wenzek et al., 2020) for training. Ablation studies motivating these technical choices are performed in Section 6. More technical details about our model are given in Appendix A.1.

### 4.1 Datasets

Contriever is trained with contrastive learning on documents sampled from a mix between Wikipedia data and CCNet data (Wenzek et al., 2020), where half the batches are sampled from each source.

First, we evaluate our model on two question answering datasets: NaturalQuestions (Kwiatkowski et al., 2019) and TriviaQA (Joshi et al., 2017). For both datasets, we use the open domain versions as introduced by Lee et al. (2019), and the English Wikipedia dump from Dec. 20, 2018 as the collection of documents to retrieve from. We report the top-k retrieval accuracy, *i.e.* the number of questions for which at least one of the top-k passages contain the answer.

Second, we use the BEIR benchmark, introduced by Thakur et al. (2021), which contains 18 retrieval datasets, corresponding to nine tasks, such as fact checking or citation prediction, and covering different domains, such as Wikipedia or scientific publications. Most datasets from BEIR do not contain a training set, and the focus of the benchmark is *zero-shot retrieval*. However, most machine learning based retrievers are still trained on supervised data, such as the large scale retrieval dataset MS MARCO (Bajaj et al., 2016). Following standard practice, we report two metrics on this benchmark: nDCG@10 and Recall@100. The nDCG@10 focuses on the ranking of the top 10 retrieved documents, and is good at evaluating rankings returned to humans, for example in a search engine. On the other hand, Recall@100 is relevant to evaluate retrievers that are used in machine learning systems, such as question answering. Indeed, such models can process hundreds of documents, and ignore their ranking (Izacard & Grave, 2020b). While nDCG@10 is the main metric of BEIR, we are more interested in the Recall@100 to evaluate bi-encoders, as our goal is to develop retrievers that can be used in ML systems. Moreover, in many settings, retrieved documents can be re-ranked with a more powerful model such as a cross-encoder, thus improving the nDCG@10.

Table 1: **Unsupervised recall@k** on the test sets of NaturalQuestions and TriviaQA. For Inverse Cloze Task and Masked Salient Spans we report the results of Sachan et al. (2021). The Masked Salient Spans model uses annotated named entity recognition data. For BM25 we report the results of Ma et al. (2021)

| | NaturalQuestions | | | TriviaQA | | |
|---|---|---|---|---|---|---|
| | R@5 | R@20 | R@100 | R@5 | R@20 | R@100 |
| Inverse Cloze Task (Sachan et al., 2021) | 32.3 | 50.9 | 66.8 | 40.2 | 57.5 | 73.6 |
| Masked salient spans (Sachan et al., 2021) | 41.7 | 59.8 | 74.9 | 53.3 | 68.2 | 79.4 |
| BM25 (Ma et al., 2021) | - | 62.9 | 78.3 | - | **76.4** | **83.2** |
| Contriever | **47.8** | **67.8** | **82.1** | **59.4** | 74.2 | **83.2** |
| *supervised model:* DPR (Karpukhin et al., 2020) | - | 78.4 | 85.4 | - | 79.4 | 85.0 |
| *supervised model:* FiD-KD (Izacard & Grave, 2020a) | 73.8 | 84.3 | 89.3 | 77.0 | 83.6 | 87.7 |

## 4.2 Baselines

First, we compare Contriever to BM25, which does not require supervision. On QA datasets, we compare to dense retrievers trained with ICT and the Masked Salient Spans from Sachan et al. (2021). On BEIR, we consider the retriever from REALM (Guu et al., 2020), and RoBERTa large fine-tuned with SimCSE (Gao et al., 2021), as unsupervised dense retrievers. We also compare to ML-based retrievers trained on MS MARCO, classified in three categories: sparse, dense and late-interaction. For sparse methods, we compare to *Splade v2* (Formal et al., 2021), which computes sparse representations of documents with BERT pre-trained model. For dense methods, we use *DPR* (Karpukhin et al., 2020) and *ANCE* (Xiong et al., 2020), which are bi-encoders trained on supervised data such as NaturalQuestions or MS MARCO. We also compare to *TAS-B* (Hofstätter et al., 2021), which performs distillation from a cross-encoder to a bi-encoder, and GenQ, which creates synthetic query-document pairs with a generative model.[1] For late-interaction, we use ColBERT (Khattab et al., 2020), which computes pairwise scores between contextualized representations of queries and documents, as well as a cross-encoder used to re-rank documents retrieved with BM25.

## 4.3 Results

First, we compare the performance of fully unsupervised models, i.e., without fine-tuning on MS MARCO or other annotated data. In Table 1, we report the retrieval performance on two question answering datasets: NaturalQuestions (Kwiatkowski et al., 2019) and TriviaQA (Joshi et al., 2017). Here, our model is competitive with a strong BM25 baseline (Ma et al., 2021), for example leading to 3 points improvement for the recall@100 on NaturalQuestions. It also outperforms previously proposed dense retrievers which were trained with ICT or salient span masking. In Figure 1 we report the recall@100 performance of unsupervised models on the BEIR benchmark. Interestingly, we observe that in this setting, Contriever is competitive compared to BM25 on all datasets, but TREC-COVID and Tóuche-2020. In particular, it obtains better performance than BM25 on 11 out of 15 datasets from the benchmark for the recall@100. Contriever also outperforms previously proposed unsupervised dense retrievers, which obtains lower performance than BM25 in general. For the nDCG@10, which puts more emphasis on the very first retrieved documents, while Contriever largely closes the gap between unsupervised retrievers and BM25, it is still outperformed by BM25 as reported in Table 11. The difference is mainly due to the fact that BM25 largely outperforms Contriever on two datasets with specific features: Trec-COVID and Tóuche-2020. Trec-COVID is an information retrieval dataset related to COVID. However data used to train Contriever were collected before the COVID outbreak, thus they may not be adapted. Tóuche-2020 contains long documents, which does not seem to be very well supported by dense neural retrievers: even after supervised training, models are still lagging behind BM25. Overall, these results show the potential of contrastive learning to train fully unsupervised dense retrievers.

---

[1]GenQ thus leads to one different model for each dataset.

Table 2: **BEIR Benchmark.** We report nDCG@10 on the test sets from the BEIR benchmark for bi-encoder methods without re-ranker. We also report the average and number of datasets where a method is the best ("Best on") over the entire BEIR benchmark (excluding three datasets because of their licence). Bold is the best overall. MS MARCO is excluded from the average. "CE" refers to cross-encoder.

|  | BM25 | BM25+CE | DPR | ANCE | TAS-B | Gen-Q | ColBERT | Splade v2 | Ours | Ours+CE |
|---|---|---|---|---|---|---|---|---|---|---|
| MS MARCO | 22.8 | 41.3 | 17.7 | 38.8 | 40.8 | 40.8 | 40.1 | 43.3 | 40.7 | **47.0** |
| Trec-COVID | 65.6 | **75.7** | 33.2 | 65.4 | 48.1 | 61.9 | 67.7 | 71.0 | 59.6 | 70.1 |
| NFCorpus | 32.5 | **35.0** | 18.9 | 23.7 | 31.9 | 31.9 | 30.5 | 33.4 | 32.8 | 34.4 |
| NQ | 32.9 | 53.3 | 47.4 | 44.6 | 46.3 | 35.8 | 52.4 | 52.1 | 49.8 | **57.7** |
| HotpotQA | 60.3 | 70.7 | 39.1 | 45.6 | 58.4 | 53.4 | 59.3 | 68.4 | 63.8 | **71.5** |
| FiQA | 23.6 | 34.7 | 11.2 | 29.5 | 30.0 | 30.8 | 31.7 | 33.6 | 32.9 | **36.7** |
| ArguAna | 31.5 | 31.1 | 17.5 | 41.5 | 42.9 | **49.3** | 23.3 | 47.9 | 44.6 | 41.3 |
| Touche-2020 | **36.7** | 27.1 | 13.1 | 24.0 | 16.2 | 18.2 | 20.2 | 36.4 | 23.0 | 29.8 |
| CQADupStack | 29.9 | 37.0. | 15.3 | 29.6 | 31.4 | 34.7 | 35.0 | - | 34.5 | **37.7** |
| Quora | 78.9 | 82.5 | 24.8 | 85.2 | 83.5 | 83.0 | 85.4 | 83.8 | **86.5** | 82.4 |
| DBPedia | 31.3 | 40.9 | 26.3 | 28.1 | 38.4 | 32.8 | 39.2 | 43.5 | 41.3 | **47.1** |
| Scidocs | 15.8 | 16.6 | 7.7 | 12.2 | 14.9 | 14.3 | 14.5 | 15.8 | 16.5 | **17.1** |
| FEVER | 75.3 | **81.9** | 56.2 | 66.9 | 70.0 | 66.9 | 77.1 | 78.6 | 75.8 | **81.9** |
| Climate-FEVER | 21.3 | 25.3 | 14.8 | 19.8 | 22.8 | 17.5 | 18.4 | 23.5 | 23.7 | **25.8** |
| Scifact | 66.5 | 68.8 | 31.8 | 50.7 | 64.3 | 64.4 | 67.1 | **69.3** | 67.7 | 69.2 |
| Avg. w/o CQA | 44.0 | 49.5 | 26.3 | 41.3 | 43.7 | 43.1 | 45.1 | 50.6 | 47.5 | 51.2 |
| Avg. | 43.0 | 48.6 | 25.5 | 40.5 | 42.8 | 42.5 | 44.4 | - | 46.6 | 50.2 |
| Best on | 1 | 3 | 0 | 0 | 0 | 1 | 0 | 1 | 1 | 9 |

Table 3: **Few-shot retrieval.** Test nDCG@10 after training on a small in-domain training set. We compare BERT and our model, with and without an intermediate fine-tuning step on MS MARCO. Note that our unsupervised pre-training alone outperforms BERT with intermediate MS MARCO fine-tuning.

|  | Additional data | SciFact | NFCorpus | FiQA |
|---|---|---|---|---|
| # queries |  | 729 | 2,590 | 5,500 |
| BM25 | - | 66.5 | 32.5 | 23.6 |
| BERT | - | 75.2 | 29.9 | 26.1 |
| Contriever | - | 84.0 | 33.6 | 36.4 |
| BERT | MS MARCO | 80.9 | 33.2 | 30.9 |
| Contriever | MS MARCO | **84.8** | **35.8** | **38.1** |

Next, we report nDCG@10 on the BEIR benchmark for different retrievers trained on MS MARCO in Table 2 (recall@100 can be found in Table 10 of appendix). We individually report results on each dataset as well as the average over 14 datasets of the BEIR Benchmark (excluding 3 for license reasons). We observe that when used as pre-training, contrastive learning leads to strong performance: contriever obtains the best results among dense bi-encoder methods for the nDCG@10, and is state-of-the-art for the recall@100 (improving the average recall@100 from 65.0 to 67.1). This strong recall@100 performance can be further exploited by using a cross-encoder[2] to re-rank the retrieved documents: this leads to the state-of-the-art on 8 datasets of the BEIR benchmark for the nDCG@10, as well as on average. It should be noted that our fine-tuning procedure on MS MARCO is simpler than for other retrievers, as we use a simple strategy for negative mining and do not use distillation. Our model would probably also benefits from improvements proposed by these retrievers, but this is beyond the scope of this paper.

Finally, we illustrate the benefit of our retriever compared to BM25 in a *few-shot* setting, where we have access to a small number of in-domain retrieval examples. This setting is common in practice, and lexical based methods, like BM25, cannot leverage the small training sets to adapt its weights. In Table 3, we report nDCG@10 on three datasets from BEIR associated with the smallest training sets, ranging from 729 to

---

[2]We use the existing `ms-marco-MiniLM-L-6-v2` cross-encoder model to perform the re-ranking.

5,500 queries. We observe that on these small datasets, our pre-training leads to better results than BERT pre-training, even when BERT is fine-tuned on MS MARCO as an intermediate step. Our pre-trained model also outperforms BM25, showing the advantage of dense retriever over lexical methods in the few-shot setting. More details are given in Appendix A.3.

## 5    Multilingual retrieval

In this section, we illustrate another advantage of learning unsupervised dense retrievers, when performing multi-lingual retrieval. While large labeled datasets are available in English, allowing to train strong dense retrievers (as shown in previous sections), this is unfortunately not the case for lower resources languages. Here, we show how unsupervised training is a promising direction. First, we show that our approach leads to strong performance, either in a full unsupervised setting, or by fine-tuning a multi-lingual model on English data such as MS MARCO. Second, we demonstrate that our model can also perform cross-lingual retrieval, by retrieving English documents from other languages queries. Unsupervised retrievers based on lexical matching, such as BM25, would obtain poor performance, especially for pairs of languages with different scripts such as English and Arabic.

### 5.1    Multilingual pre-training

Our multilingual model, called *mContriever*, is jointly pre-trained on 29 languages. The multilingual pre-training largely follows the method discussed in previous sections, but it differs by few particularities related to the pre-training data and the hyperparameters used. The model is initialized with the multilingual version of BERT, mBERT, trained on 104 languages. For the pre-training data, we consider the languages contained in CCNet (Wenzek et al., 2020) that are also part of our evaluation datasets. This results in a training set containing the CCNet data for 29 languages detailed in Table 12. During pre-training, examples are uniformly sampled over languages, i.e. the probability that a training sample comes from a specific language is the same for all languages. We observed that increasing the number of languages considered for pre-training generally deteriorates performance as reported in Appendix B.3 similarly to what has been observed for multilingual masked language models (Conneau et al., 2019). We pre-trained our multilingual mContriever with a queue size of 32768. This generally improves stability, and is able to compensate for the additional instabilities observed in the multilingual setting. More detailed hyperparameters are given in Appendix B.1.

### 5.2    Fine-tuning

Large labeled datasets for information retrieval are generally available only in English. It is therefore natural to investigate whether large monolingual datasets can be leveraged for multilingual retrieval. We consider fine-tuning our pre-trained mContriever model on MS MARCO. This generally improves performance in all languages. The model trained on MS MARCO can be further fine-tuned on Mr. TyDi achieving state-of-the-art performance on this dataset. Further details regarding fine-tuning are given in Appendix B.2.

### 5.3    Evaluation

We evaluate the performance of our pre-trained model with and without fine-tuning on English data on two different benchmarks. First, we consider Mr. TyDi (Zhang et al., 2021), a multilingual information retrieval benchmark derived from TyDi QA (Clark et al., 2020). Given a question, the goal is to find relevant documents in a pool of Wikipedia documents in the same language.

In Mr. TyDi the pool of documents is restricted to the language of the query. Being able to retrieve relevant documents from another language is desirable to leverage large source of information that may no be available in all languages. In order to evaluate cross-lingual retrieval performance we derive an evaluation setting from MKQA (Longpre et al., 2020). Given a question in a specific language, we perform retrieval in English Wikipedia, and evaluate if the English answer is in the retrieved documents. The MKQA dataset makes this possible by providing the same questions and answers in 26 languages. We remove unanswerable questions, questions accepting a binary yes/no answer and questions with long answers from the original MKQA dataset.

Table 4: **Multilingual retrieval on Mr. TyDi.** We report MRR@100 and Recall@100 on the test sets of Mr. TyDi. mContriever fine-tuned on MS MARCO is compared against its counterparts without contrastive pre-training using a similar fine-tuning recipe, referred to as *mBERT + MS MARCO*, as well as a model initialized with XLM-R referred to as *XLM-R + MS MARCO*. We also report the results after fine-tuning on Mr. TyDi for the model trained on MS MARCO.

| | ar | bn | en | fi | id | ja | ko | ru | sw | te | th | avg |
|---|---|---|---|---|---|---|---|---|---|---|---|---|
| | | | | | | MRR@100 | | | | | | |
| BM25 (Zhang et al., 2021) | 36.7 | 41.3 | 15.1 | 28.8 | 38.2 | 21.7 | 28.1 | 32.9 | 39.6 | 42.4 | 41.7 | 33.3 |
| mDPR (Zhang et al., 2021) | 26.0 | 25.8 | 16.2 | 11.3 | 14.6 | 18.1 | 21.9 | 18.5 | 7.3 | 10.6 | 13.5 | 16.7 |
| Hybrid (Zhang et al., 2021) | 49.1 | 53.5 | 28.4 | 36.5 | 45.5 | 35.5 | 36.2 | 42.7 | 40.5 | 42.0 | 49.2 | 41.7 |
| mBERT + MS MARCO | 34.8 | 35.1 | 25.7 | 29.6 | 36.3 | 27.1 | 28.1 | 30.0 | 37.4 | 39.6 | 20.3 | 31.3 |
| XLM-R + MS MARCO | 36.5 | 41.7 | 23.0 | 32.7 | 39.2 | 24.8 | 32.2 | 29.3 | 35.1 | 54.7 | 38.5 | 35.2 |
| mContriever | 27.3 | 36.3 | 9.2 | 21.1 | 23.5 | 19.5 | 22.3 | 17.5 | 38.3 | 22.5 | 37.2 | 25.0 |
| + MS MARCO | 43.4 | 42.3 | 27.1 | 35.1 | 42.6 | 32.4 | 34.2 | 36.1 | 51.2 | 37.4 | 40.2 | 38.4 |
| + Mr. Tydi | **72.4** | **67.2** | **56.6** | **60.2** | **63.0** | **54.9** | **55.3** | **59.7** | **70.7** | **90.3** | **67.3** | **65.2** |
| | | | | | | Recall@100 | | | | | | |
| BM25 (Zhang et al., 2021) | 80.0 | 87.4 | 55.1 | 72.5 | 84.6 | 65.6 | 79.7 | 66.0 | 76.4 | 81.3 | 85.3 | 74.3 |
| mDPR (Zhang et al., 2021) | 62.0 | 67.1 | 47.5 | 37.5 | 46.6 | 53.5 | 49.0 | 49.8 | 26.4 | 35.2 | 45.5 | 47.3 |
| Hybrid (Zhang et al., 2021) | 86.3 | 93.7 | 69.6 | 78.8 | 88.7 | 77.8 | 70.6 | 76.0 | 78.6 | 82.7 | 87.5 | 80.9 |
| mBERT + MS MARCO | 81.1 | 88.7 | 77.8 | 74.2 | 81.0 | 76.1 | 66.7 | 77.6 | 74.1 | 89.5 | 57.8 | 76.8 |
| XLM-R + MS MARCO | 79.9 | 84.2 | 73.1 | 81.6 | 87.4 | 70.9 | 71.1 | 74.1 | 73.9 | 91.2 | 89.5 | 79.7 |
| mContriever | 82.0 | 89.6 | 48.8 | 79.6 | 81.4 | 72.8 | 66.2 | 68.5 | 88.7 | 80.8 | 90.3 | 77.2 |
| + MS MARCO | 88.7 | 91.4 | 77.2 | 88.1 | 89.8 | 81.7 | 78.2 | 83.8 | 91.4 | 96.6 | 90.5 | 87.0 |
| + Mr. Tydi | **94.0** | **98.6** | **92.2** | **92.7** | **94.5** | **88.8** | **88.9** | **92.4** | **93.7** | **98.9** | **95.2** | **93.6** |

This results in an evaluation set of 6619 queries. It should be noted that methods based on term matching such as BM25 are intrinsically limited in this cross-lingual retrieval setting because similar terms in different languages may not match.

## 5.4 Baselines

On Mr. TyDi (Zhang et al., 2021) we report results from the original paper. This includes a BM25 baseline, a model fine-tuned on NaturalQuestions using the DPR pipeline, and an hybrid model combining the two.

On our cross-lingual evaluation benchmark derived from MKQA, we consider the retriever of the CORA question answering pipeline (Asai et al., 2021), trained on a combination of datasets containing the English NaturalQuestions and the cross-lingual XOR-TyDi QA, with data augmentation based on translation.

Additionally, to isolate the effect of contrastive pre-training, we also compare mContriever fine-tuned on MS MARCO to its counterparts without contrastive pre-training, initialized from mBERT. This model is referred as *mBERT + MSMARCO* in tables. We also report results obtained by fine-tuning XLM-R (Conneau et al., 2019) on MS-MARCO. For both models we use the same hyper-parameters used to fine-tune mContriever on MS MARCO except for the temperature, additional details are reported in Appendix B.2.

## 5.5 Results

We report results on Mr. TyDi in Table 4. The effectiveness of the multilingual pre-training appears clearly as the pre-trained model fine-tuned on MS MARCO achieve significantly better performance than its counterparts without pre-training when fine-tuned using the same pipeline. Interestingly fine-tuning on these English-only datasets improves performance on all languages. Moreover our unsupervised mContriever outperforms BM25 for the Recall@100, and after fine-tuning on MS MARCO it achieves state-of-the-art performance for this metric. Performance can be further improved by fine-tuning on the train set of Mr. TyDi. This appears to be

Table 5: **Cross-lingual retrieval on MKQA.** We report the average on all languages included in MKQA for the Recall@100 and the Recall@20, and report the Recall@100 for a subset of languages. Complete results are reported in Table 13 and Table 14 of appendix.

|  | Avg. R@20 | Avg. R@100 | en | ar | ja | ko | es | he | de |
|---|---|---|---|---|---|---|---|---|---|
| CORA (Asai et al., 2021) | 49.0 | 59.8 | **75.6** | 44.5 | 47.0 | 45.5 | 69.2 | 48.3 | **68.1** |
| mBERT + MS MARCO | 45.3 | 57.9 | 74.2 | 44.0 | 51.7 | 48.2 | 63.9 | 46.8 | 59.6 |
| XLM-R + MS MARCO | 46.9 | 59.6 | 73.4 | 42.5 | 53.2 | 49.6 | 63.4 | 46.9 | 61.1 |
| mContriever | 31.4 | 49.2 | 65.3 | 43.0 | 47.1 | 44.8 | 37.2 | 44.7 | 49.0 |
| + MS MARCO | **53.9** | **65.6** | **75.6** | **53.3** | **60.4** | **55.4** | **70.0** | **59.6** | 66.6 |

Table 6: **MoCo vs. in-batch negatives.** In this table, we report nDCG@10 on the BEIR benchmark for in-batch negatives and MoCo, without fine-tuning on the MS MARCO dataset.

|  | NFCorpus | NQ | FiQA | ArguAna | Quora | DBPedia | SciDocs | FEVER | AVG |
|---|---|---|---|---|---|---|---|---|---|
| MoCo | 26.2 | 13.1 | 13.7 | 33.0 | 69.5 | 20.0 | 11.9 | 57.6 | 30.1 |
| In-batch negatives | 24.2 | 21.6 | 13.0 | 33.7 | 74.9 | 17.9 | 13.6 | 56.1 | 31.9 |

particularly important for the MRR@100 metric which put emphasis on the quality of the first documents retrieved. For this metric our unsupervised model is still lagging behind BM25.

Results on the cross-lingual retrieval benchmark derived from MKQA are reported in Table 5, with per language details for the Recall@100 and Recall@20 reported in Table 13 and Table 14 of appendix. Interestingly using only supervised training data in English, our mContriever fine-tuned on MS MARCO outperforms the CORA retriever. Also, similarly to the results reported on Mr. TyDi, adding multilingual contrastive pre-training before fine-tuning on MS MARCO improves performance over its counterpart without pre-training. On MKQA, evaluation is performed by lowercasing both queries and documents, we observed that this improves performance. This does not impact the CORA retriever which is based on an uncased version of mBERT.

## 6 Ablation studies

In this section, we investigate the influence of different design choices on our method. In these ablations, all the models are pre-trained on English Wikipedia for 200k gradient steps, with a batch size of 2,048 (on 32 GPUs). Each fine-tuning on MS MARCO takes 20k gradient steps with a batch size of 512 (on 8 GPUs), using AdamW and no hard negatives.

**MoCo vs. in-batch negatives.** First, we compare the two contrastive pre-training methods: MoCo and in-batch negatives. As in in-batch negatives, the number of negative examples is equal to the batch size, we train models with a batch size of 4,096 and restrict the queue in MoCo to the same number of elements. This experiment measures the effect of using of momentum encoder for the keys instead of the same network as for the queries. Using a momentum also prevents from backpropagating the gradient through the keys. We report results, without fine-tuning on MS MARCO in Table 6. We observe that the difference of performance between the two methods is small, especially after fine-tuning on MS MARCO. We thus propose to use MoCo as our contrastive learning framework, since it scales to a larger number of negative examples without the need to increase the batch size.

**Number of negative examples.** Next, we study the influence of the number of negatives used in the contrastive loss, by varying the queue size of the MoCo algorithm. We consider values ranging from 2,048 to 131,072, and report results in Figure 2. We see that on average over the BEIR benchmark, increasing the number of negatives leads to better retrieval performance, especially in the unsupervised setting. However, we note that this effect is not equally strong for all datasets.

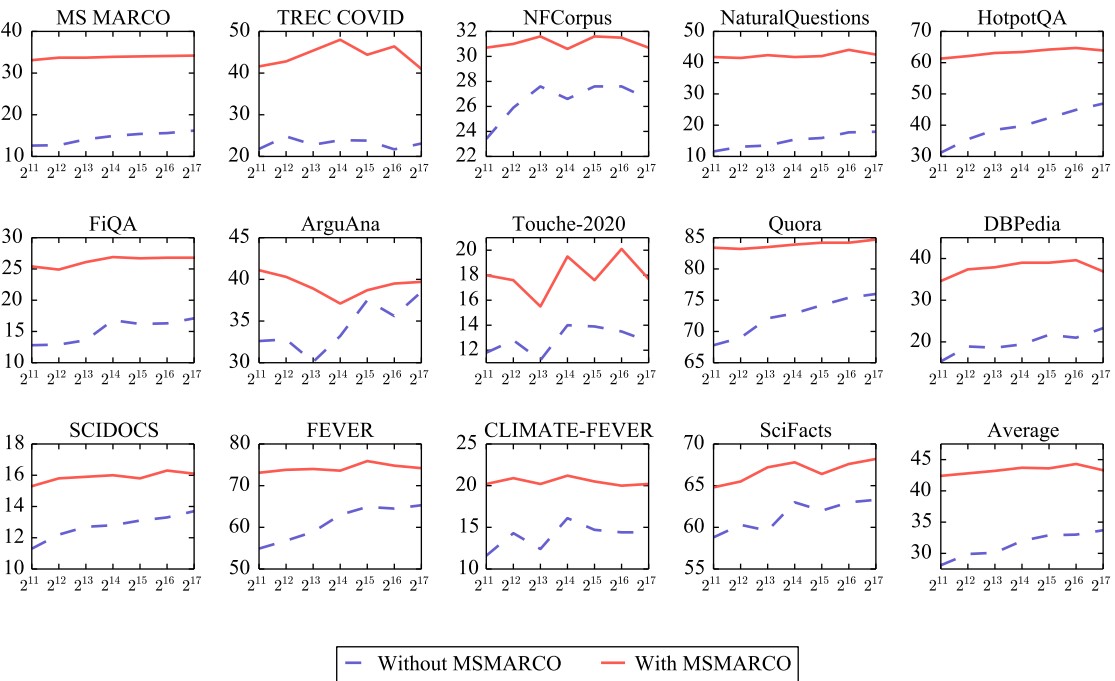

Figure 2: **Impact of the number of negatives.** We report nDCG@10 as a function of the queue size, with and without fine-tuning on MS MARCO. We report numbers using the MoCo framework where the keys for the negatives are computed with the momentum encoder and stored in a queue.

**Data augmentations.** Third, we compare different ways to generate pairs of positive examples from a single document or chunk of text. In particular, we compare random cropping, which leads to pairs with overlap, and the inverse cloze task, which was previously considered to pre-train retrievers. Interestingly, as shown in Table 7, the random cropping strategy outperforms the inverse cloze task in our setting. We believe that random cropping, leading to the identical distributions of keys and queries, leads to more stable training with MoCo compared to ICT. This might explains part of the difference of performance between the two methods. We also investigate whether additional data perturbations, such as random word deletion or replacement, are beneficial for retrieval.

**Training data.** Finally, we study the impact of the pre-training data on the performance of our retriever, by training on Wikipedia, CCNet or a mix of both sources of data. We report results in Table 8, and observe that there is no clear winner between the two data sources. Unsurprisingly, training on the more diverse CCNet data leads to strong improvements on datasets from different domains than Wikipedia, such as FiQA or Quora. On the other hand, on a dataset like FEVER, training on Wikipedia leads to better results. To get the best of both worlds, we consider two strategies to mix the two data sources. In the "50/50%" strategy, examples are sampled uniformly across domain, meaning that half the batches are from Wikipedia and the other half from CCNet. In the "uniform" strategy, examples are sampled uniformly over the union of the dataset. Since CCNet is significantly larger than Wikipedia, this means that most of the batches are from CCNet.

**Impact of fine-tuning on MS MARCO.** To isolate the impact of pre-training from the impact of fine-tuning on MS MARCO, we apply the same fine-tuning to the BERT base uncased model. We report results in Table 9, and observe that when applied to BERT, our fine-tuning leads to results that are lower than the state-of-the-art. Hence, we believe that most of the improvement compared to the state-of-the-art retrievers can be attributed to our contrastive pre-training strategy.

Table 7: **Impact of data augmentations.** We report nDCG@10 without fine-tuning on MS MARCO.

|  | NFCorpus | NQ | ArguAna | Quora | DBPedia | SciDocs | FEVER | Overall |
|---|---|---|---|---|---|---|---|---|
| ICT | 23.2 | 19.4 | 31.6 | 27.6 | 21.3 | 10.6 | 55.6 | 25.9 |
| Crop | 27.6 | 17.7 | 35.6 | 75.4 | 21.0 | 13.3 | 64.5 | 32.2 |
| Crop + delete | 26.8 | 20.8 | 35.8 | 77.3 | 21.5 | 14.0 | 67.9 | 33.8 |
| Crop + replace | 27.7 | 18.7 | 36.2 | 75.6 | 22.0 | 13.0 | 66.8 | 32.9 |

Table 8: **Training data.** We report nDCG@10 without fine-tuning on MS MARCO.

|  | NFCorpus | NQ | FiQA | ArguAna | Quora | DBPedia | SciDocs | FEVER | Overall |
|---|---|---|---|---|---|---|---|---|---|
| Wiki | 27.6 | 17.7 | 16.3 | 35.6 | 75.4 | 21.0 | 13.3 | 64.5 | 33.0 |
| CCNet | 29.5 | 25.8 | 26.2 | 35.2 | 80.6 | 20.5 | 14.9 | 60.9 | 34.9 |
| Uniform | 31.0 | 19.4 | 25.1 | 37.8 | 80.4 | 21.5 | 14.7 | 59.8 | 33.9 |
| 50/50% | 31.5 | 18.6 | 23.3 | 36.2 | 79.1 | 22.1 | 13.7 | 64.1 | 34.7 |

Table 9: **Fine-tuning.** We report nDCG@10 after fine-tuning BERT and our model on MS MARCO.

|  | NFCorpus | NQ | FiQA | ArguAna | Quora | DBPedia | SciDocs | FEVER | Overall |
|---|---|---|---|---|---|---|---|---|---|
| BERT | 28.2 | 44.6 | 25.9 | 35.0 | 84.0 | 34.4 | 13.0 | 69.8 | 42.0 |
| Contriever | 33.2 | 50.2 | 28.8 | 46.0 | 85.4 | 38.8 | 16.0 | 77.7 | 46.5 |

## 7 Discussion

In this work, we propose to explore the limits of contrastive pre-training to learn dense text retrievers. We use the MoCo technique, which allows to train models with a large number of negative examples. We make several interesting observations: first, we show that neural networks trained without supervision using contrastive learning exhibit good retrieval performance, which are competitive with BM25 (albeit not state-of-the-art). These results can be further improved by fine-tuning on the supervised MS MARCO dataset, leading to strong results, in particular for recall@100. Based on that observation, we use a cross-encoder to re-rank documents retrieved with our model, leading to new state-of-the-art on the competitive BEIR benchmark. We also performed extensive ablation experiments, and observed that independent random cropping seems to be a strong alternative to the inverse Cloze task for training retrievers.

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

## A    Technical details for Contriever

### A.1    Contrastive pre-training

For the model with fine-tuning on MS MARCO, we use the MoCo algorithm He et al. (2020) with a queue of size 131,072, a momentum value of 0.9995 and a temperature of 0.05. We use the random cropping data augmentation, with documents of 256 tokens and span sizes sampled between 5% and 50% of the document length. Documents are simply random piece of text sampled from a mix between Wikipedia and CCNet data (Wenzek et al., 2020), where half the batches are sampled from each source. We also apply token deletion with a probability of 10%. We optimize the model with the AdamW (Loshchilov & Hutter, 2019) optimizer, with learning rate of $5 \cdot 10^{-5}$, batch size of 2,048 and 500,000 steps. We initialize the network with the publicly available BERT base uncased model.

### A.2    Fine-tuning on MS MARCO

For the fine-tuning on MS MARCO we do not use the MoCo algorithm and simply use in-batch negatives. We use the ASAM optimizer (Kwon et al., 2021), with a learning rate of $10^{-5}$ and a batch size of 1024 with a temperature of 0.05, also used during pre-training. We train an initial model with random negative examples for 20000 steps, mine hard negatives with this first model, and re-train a second model with those. Each query is associated with a gold document and a negative document, which is a random document in the first phase and a hard negative 10% of the time in the second phase. For each query, all documents from the current batch aside of the gold document are used as negatives.

### A.3    Few-shot training

For the few-shot evaluation presented in Table 3, we train for 500 epochs on each dataset with a batch size of 256 with in-batch random negatives. We evaluate performance performance on the development set every 100 gradient updates and perform early stopping based on this metric. For SciFact, we hold out randomly 10% of the training data and use them as development set, leading to a train set containing 729 samples.

Table 10: **BEIR Benchmark.** We report the recall@100 on the test sets from the BEIR benchmark for bi-encoder methods. We report the capped recall@100 on Trec-COVID following the original BEIR setup. Note that using a cross-encoder to re-rank the top-100 documents do not change the recall@100, hence, we do not include these methods in this table. We also report the average and number of datasets where a method is the best ("Best on") over the entire BEIR benchmark (excluding three datasets because of their licence). Bold is the best overall. On Trec-COVID we report the capped Recall@100, see Thakur et al. (2021) for more details. MS MARCO is excluded from the average.

|  | BM25 | DPR | ANCE | TAS-B | Gen-Q | ColBERT | Splade v2 | Ours |
|---|---|---|---|---|---|---|---|---|
| MS MARCO | 65.8 | 55.2 | 85.2 | 88.4 | 88.4 | 86.5 | - | **89.1** |
| Trec-COVID | **49.8** | 21.2 | 45.7 | 38.7 | 45.6 | 46.4 | 12.3 | 40.7 |
| NFCorpus | 25.0 | 20.8 | 23.2 | 28.0 | 28.0 | 25.4 | 27.7 | **30.0** |
| NQ | 76.0 | 88.0 | 83.6 | 90.3 | 86.2 | 91.2 | **93.0** | 92.5 |
| HotpotQA | 74.0 | 59.1 | 57.8 | 72.8 | 67.3 | 74.8 | **82.0** | 77.7 |
| FiQA | 53.9 | 34.2 | 58.1 | 59.3 | 61.8 | 60.3 | 62.1 | **65.6** |
| ArguAna | 94.2 | 75.1 | 93.7 | 94.2 | **97.8** | 91.4 | 97.2 | 97.7 |
| Touche-2020 | **53.8** | 30.1 | 45.8 | 43.1 | 45.1 | 43.9 | 35.4 | 29.4 |
| CQADupStack | 60.6 | 40.3 | 57.9 | 62.2 | 65.4 | 62.4 | - | **66.3** |
| Quora | 97.3 | 47.0 | 98.7 | 98.6 | 98.8 | 98.9 | 98.7 | **99.3** |
| DBPedia | 39.8 | 34.9 | 31.9 | 49.9 | 43.3 | 46.1 | **57.5** | 54.1 |
| Scidocs | 35.6 | 21.9 | 26.9 | 33.5 | 33.2 | 34.4 | 36.4 | **37.8** |
| Fever | 93.1 | 84.0 | 90.0 | 93.7 | 92.8 | 93.4 | **95.1** | 94.9 |
| Climate-fever | 43.6 | 39.0 | 44.5 | 53.4 | 45.0 | 44.4 | 52.4 | **57.4** |
| Scifact | 90.8 | 72.7 | 81.6 | 89.1 | 89.3 | 87.8 | 92.0 | **94.7** |
| Avg. w/o CQA | 63.6 | 48.3 | 60.1 | 65.0 | 64.2 | 64.5 | 64.8 | 67.1 |
| Avg. | 63.4 | 47.7 | 60.0 | 64.8 | 64.2 | 64.3 | - | 67.0 |
| Best on | 2 | 0 | 0 | 0 | 1 | 0 | 4 | 7 |

# B  Multilingual retrieval with mContriever

## B.1  Hyperparameters for multilingual contrastive pre-training

The pre-trained mContriever model is pre-trained for 500,000 steps with a queue of size 32768, and temperature of 0.05 and a momentum value of 0.999. We optimize the model with the AdamW (Loshchilov & Hutter, 2019) optimizer, with learning rate of $5 \cdot 10^{-5}$. The learning rate follows a linear warmup for 20,000 steps followed by linear decay until the end of training. Languages used for pre-training are detailed in Table 12.

## B.2  Hyperparameters for multilingual fine-tuning

We fine-tune mContriever using in-batch negatives, AdamW optimizer (Loshchilov & Hutter, 2019), a learning rate of $10^{-5}$, and a batch size of 1024 samples with a temperature $\tau$ of 0.05. On MS MARCO and Mr. TyDi the model is trained for 20k gradient steps. We notice overfitting on NaturalQuestions, and thus reduced the training to 1k gradient steps. We use a warmup of 1000 gradient steps with linear decay afterwards in all cases. Hard negatives are mined on Mr. TyDi with the model trained on MS MARCO. We did not observe significant improvements using hard negatives on MS MARCO and NaturalQuestions.

For the fine-tuning on MS MARCO of the models initiliazed from mBERT (resp. XLM-R) without contrastive pre-training, we use a temperature $\tau$ of 1 (resp. 5). We tried temperatures in $\{10, 5, 2, 1, 0.1, 0.05\}$ and chose the one leading to the best performance. We observed a decrease in performance for lower temperatures. We fine-tuned mContriever with $\tau = 0.05$ following the temperature used during pre-training. We followed the temperature $\tau = 0.05$ used for the training of Contriever, and did not test other temperatures for the contrastive pre-training of the multilingual model, mContriever.

Table 11: **Unsupervised retrieval.** Performance of unsupervised methods on the BEIR datasets. We report the capped recall@100 on Trec-COVID following the original BEIR setup. For SimCSE we report results of the model using RoBERTa large. REALM uses annotated entity recognition data for training. On Trec-COVID we report the capped Recall@100, see Thakur et al. (2021) for more details.

| Model (→) | BM25 | BERT | SimCSE | REALM | Contriever |
|---|---|---|---|---|---|
| Dataset (↓) | | | Recall@100 | | |
| MS MARCO | 65.8 | 3.5 | 33.6 | 52.6 | **67.2** |
| Trec-COVID | **49.8** | 10.6 | 26.8 | 8.1 | 17.2 |
| NFCorpus | 25.0 | 6.7 | 18.2 | 23.0 | **29.4** |
| NQ | 76.0 | 14.3 | 42.9 | 58.1 | **77.1** |
| HotpotQA | **74.0** | 15.8 | 42.7 | 56.1 | 70.4 |
| FiQA-2018 | 53.9 | 6.9 | 41.0 | 28.0 | **56.2** |
| ArguAna | 94.2 | 59.1 | **95.2** | 73.1 | 90.1 |
| Tóuche-2020 | **53.8** | 3.0 | 18.6 | 11.5 | 22.5 |
| CQADupStack | 60.6 | 11.0 | 48.9 | 35.5 | **61.4** |
| Quora | 97.3 | 74.6 | 97.9 | 92.7 | **98.7** |
| DBPedia | 39.8 | 7.1 | 21.5 | 33.0 | **45.3** |
| SCIDOCS | 35.6 | 11.3 | 23.0 | 23.1 | **36.0** |
| Fever | 93.1 | 13.6 | 50.8 | 82.6 | **93.6** |
| Climate-fever | 43.6 | 12.8 | **44.8** | 42.3 | 44.1 |
| SciFact | 90.8 | 35.2 | 75.3 | 83.8 | **92.6** |
| Avg. | 63.6 | 19.0 | 45.4 | 46.9 | 60.1 |
| Best on | 3 | 0 | 2 | 0 | 10 |
| | | | NDCG@10 | | |
| MS MARCO | **22.8** | 0.6 | 8.8 | 15.2 | 20.6 |
| Trec-COVID | **65.6** | 16.6 | 38.6 | 20.1 | 27.4 |
| NFCorpus | **32.5** | 2.5 | 14.0 | 24.1 | 31.7 |
| NQ | **32.9** | 2.7 | 12.6 | 15.2 | 25.4 |
| HotpotQA | **60.3** | 4.9 | 23.3 | 40.5 | 48.1 |
| FiQA-2018 | 23.6 | 1.4 | 14.8 | 9.7 | **24.5** |
| ArguAna | 31.5 | 23.1 | **45.6** | 22.8 | 37.9 |
| Tóuche-2020 | **36.7** | 3.4 | 11.6 | 7.3 | 19.3 |
| CQADupStack | **29.9** | 2.5 | 20.2 | 13.5 | 28.4 |
| Quora | 78.9 | 3.9 | 81.5 | 71.6 | **83.5** |
| DBPedia | **31.3** | 3.9 | 13.7 | 22.7 | 29.2 |
| SCIDOCS | **15.8** | 2.7 | 7.4 | 9.0 | 14.9 |
| FEVER | **75.3** | 4.9 | 20.1 | 42.9 | 68.2 |
| Climate-fever | **21.3** | 4.1 | 17.6 | 14.3 | 15.5 |
| SciFact | **66.5** | 9.8 | 38.5 | 47.1 | 64.9 |
| Avg. | 41.7 | 8.7 | 24.6 | 25.1 | 36.0 |
| Best on | 12 | 0 | 1 | 0 | 2 |

## B.3 Curse of multilinguality

We tried to pre-train models on different sets of languages. We generally observed performance deterioration when scaling to more languages similarly to what has been observed for general multilingual masked language models Conneau et al. (2019). In Table 15 we report results on Mr. TyDi with a model pre-trained on the 11 languages of Mr. TyDi versus the model used in the rest of the paper which has been pre-trained on 29 languages including the 11 languages of Mr. TyDi as detailed in Table 12. We also report performance of these models after training on MS MARCO, eventually followed by further fine-tuning on Mr. TyDi. It appears that the performance of the unsupervised model and the performance after fine-tuning on MS MARCO are better for the model pre-trained only on 11 languages. The difference is mitigated after fine-tuning on Mr. TyDi.

Table 12: **List of languages used for multilingual retrieval.**

| | ar | bn | da | de | en | es |
|---|---|---|---|---|---|---|
| Language | Arabic | Bengali | Danish | German | English | Spanish |
| Pre-training | ✓ | ✓ | ✓ | ✓ | ✓ | ✓ |
| Mr. TyDi | ✓ | ✓ | ✗ | ✗ | ✗ | ✓ |
| MKQA | ✓ | ✗ | ✓ | ✓ | ✓ | ✓ |
| | fi | fr | he | hu | it | id |
| Language | Finnish | French | Hebrew | Hungarian | Italian | Indonesian |
| Pre-training | ✓ | ✓ | ✓ | ✓ | ✓ | ✓ |
| Mr. TyDi | ✓ | ✗ | ✗ | ✗ | ✗ | ✓ |
| MKQA | ✓ | ✓ | ✓ | ✓ | ✓ | ✗ |
| | ja | km | ko | ms | nl | no |
| Language | Japanese | Khmer | Korean | Malay | Dutch | Norwegian |
| Pre-training | ✓ | ✓ | ✓ | ✓ | ✓ | ✓ |
| Mr. TyDi | ✓ | ✗ | ✓ | ✗ | ✗ | ✗ |
| MKQA | ✓ | ✓ | ✓ | ✓ | ✓ | ✓ |
| | pl | pt | ru | sv | sw | te |
| Language | Polish | Portugese | Russian | Swedish | Swahili | Telugu |
| Pre-training | ✓ | ✓ | ✓ | ✓ | ✓ | ✓ |
| Mr. TyDi | ✗ | ✗ | ✓ | ✗ | ✓ | ✓ |
| MKQA | ✓ | ✓ | ✓ | ✓ | ✗ | ✗ |
| | th | tr | vi | zh-cn | zh-hk | zh-tw |
| Language | Thai | Turkish | Vietnamese | Chinese (Simplified) | Chinese (Hong Kong) | Chinese (Traditional) |
| Pre-training | ✓ | ✓ | ✓ | ✓ | ✗ | ✓ |
| Mr. TyDi | ✓ | ✗ | ✗ | ✗ | ✗ | ✗ |
| MKQA | ✓ | ✓ | ✓ | ✓ | ✓ | ✓ |

Table 13: **Recall@100 on MKQA for cross-lingual retrieval** in the setting described in Section 5.3.

| | avg | en | ar | fi | ja | ko | ru | es | sv | he | th | da | de | fr |
|---|---|---|---|---|---|---|---|---|---|---|---|---|---|---|
| CORA | 59.8 | **75.6** | 44.5 | 61.3 | 47.0 | 45.5 | 58.6 | 69.2 | 68.0 | 48.3 | 44.4 | 68.9 | **68.1** | **70.2** |
| mBERT + MS MARCO | 57.9 | 74.2 | 44.0 | 51.7 | 55.7 | 48.2 | 57.4 | 63.9 | 62.7 | 46.8 | 51.7 | 63.7 | 59.6 | 65.2 |
| XLM-R + MS MARCO | 59.2 | 73.4 | 42.4 | 57.7 | 53.1 | 48.6 | 58.5 | 62.9 | 67.5 | 46.9 | 61.5 | 66.9 | 60.9 | 62.4 |
| Contriever | 49.2 | 65.3 | 43.0 | 43.1 | 47.1 | 44.8 | 51.8 | 37.2 | 54.5 | 44.7 | 51.4 | 49.3 | 49.0 | 50.2 |
| + MS MARCO | **65.6** | 75.6 | **53.3** | **66.6** | 60.4 | 55.4 | 64.7 | 70.0 | 70.8 | 59.6 | 63.5 | **72.0** | 66.6 | 70.1 |
| | it | nl | pl | pt | hu | vi | ms | km | no | tr | zh-cn | zh-hk | zh-tw | |
| CORA | 68.3 | **72.0** | 65.6 | 67.9 | 59.5 | 61.2 | 67.9 | 35.6 | 68.3 | 61.5 | 52.0 | 52.8 | 52.8 | |
| mBERT + MS MARCO | 64.1 | 66.7 | 59.0 | 61.9 | 57.5 | 58.6 | 62.8 | 32.9 | 63.2 | 56.0 | 58.4 | 59.3 | 59.3 | |
| XLM-R + MS MARCO | 58.1 | 66.4 | 61.0 | 62.0 | 60.1 | 62.4 | 66.1 | **46.6** | 65.9 | 60.6 | 55.8 | 55.5 | 55.7 | |
| Contriever | 56.7 | 61.7 | 44.4 | 54.5 | 47.7 | 45.1 | 56.7 | 27.8 | 50.2 | 44.3 | 54.3 | 51.9 | 52.5 | |
| + MS MARCO | **70.3** | 71.4 | **68.8** | **68.5** | **66.7** | **67.8** | **71.6** | 37.8 | **71.5** | **68.7** | **64.1** | **64.5** | **64.3** | |

Table 14: **Recall@20 on MKQA for cross-lingual retrieval** in the setting described in Section 5.3.

| | avg | en | ar | fi | ja | ko | ru | es | sv | he | th | da | de | fr |
|---|---|---|---|---|---|---|---|---|---|---|---|---|---|---|
| CORA | 49.0 | **68.5** | 31.7 | 49.7 | 34.1 | 33.1 | 46.5 | **60.3** | 58.1 | 36.8 | 33.6 | **59.4** | **58.5** | 61.6 |
| mBERT + MS MARCO | 45.3 | 65.5 | 30.2 | 38.9 | 41.7 | 34.5 | 44.3 | 52.4 | 50.5 | 32.6 | 38.5 | 52.5 | 46.6 | 53.8 |
| XLM-R + MS MARCO | 46.7 | 64.5 | 29.0 | 45.1 | 39.7 | 34.9 | 45.9 | 51.4 | 56.1 | 32.5 | 49.4 | 55.8 | 48.3 | 50.5 |
| Contriever | 31.4 | 50.2 | 26.6 | 26.7 | 29.4 | 27.9 | 32.7 | 20.7 | 37.6 | 22.2 | 31.1 | 31.2 | 31.2 | 30.7 |
| + MS MARCO | **53.9** | 67.2 | **40.1** | **55.1** | **46.2** | **41.7** | **52.3** | 59.3 | **60.0** | **45.6** | **52.0** | 62.0 | 54.8 | 59.3 |
| | it | nl | pl | pt | hu | vi | ms | km | no | tr | zh-cn | zh-hk | zh-tw | |
| CORA | 58.2 | **63.5** | 54.3 | **58.4** | 47.6 | 49.8 | 57.6 | 24.8 | 58.8 | 49.1 | 38.6 | 40.5 | 39.6 | |
| mBERT + MS MARCO | 52.1 | 55.3 | 45.6 | 49.5 | 44.6 | 46.9 | 49.9 | 21.5 | 51.3 | 42.7 | 44.6 | 45.3 | 45.5 | |
| XLM-R + MS MARCO | 45.4 | 54.5 | 48.5 | 49.6 | 47.3 | 49.7 | 54.0 | **33.4** | 53.7 | 48.7 | 42.4 | 42.4 | 42.0 | |
| Contriever | 38.6 | 45.1 | 25.1 | 37.6 | 28.3 | 27.3 | 39.6 | 15.7 | 33.2 | 26.5 | 35.0 | 32.7 | 32.5 | |
| + MS MARCO | **59.4** | 60.9 | **58.1** | 56.9 | **55.2** | **55.9** | **60.9** | 26.2 | **61.0** | 56.7 | 50.9 | 51.9 | 51.2 | |

Table 15: **Performance dilution for multilingual retrievers.** We report MRR@100 and R@100 on the test set of Mr. TyDi after pre-training on two different sets of languages, one containing the 11 languages of Mr. TyDi which is included in the set of 29 languages used to train mContriever used in the rest of the paper. We also report results of these models after fine-tuning on MS MARCO, potentially followed by a final fine-tuning stage on Mr. TyDi.

| | ar | bn | en | fi | id | ja | ko | ru | sw | te | th | avg |
|---|---|---|---|---|---|---|---|---|---|---|---|---|
| | | | | | | MRR@100 | | | | | | |
| 11 languages | 28.9 | 38.6 | 10.2 | 22.9 | 26.2 | 21.7 | 26.3 | 20.5 | 39.0 | 20.2 | 40.6 | 26.8 |
| 29 languages | 27.3 | 36.3 | 9.2 | 21.1 | 23.5 | 19.5 | 22.3 | 17.5 | 38.3 | 22.5 | 37.2 | 25.0 |
| *+ MS MARCO* | | | | | | | | | | | | |
| 11 languages | 44.0 | 41.1 | 26.8 | 38.3 | 43.4 | 34.7 | 37.1 | 37.6 | 55.3 | 32.1 | 45.8 | 39.7 |
| 29 languages | 43.4 | 42.3 | 27.1 | 35.1 | 42.6 | 32.4 | 34.2 | 36.1 | 51.2 | 37.4 | 40.2 | 38.4 |
| *+ Mr. TyDi* | | | | | | | | | | | | |
| 11 languages | 73.5 | 66.8 | 55.9 | 60.4 | 62.7 | 53.8 | 55.8 | 60.6 | 69.5 | 89.8 | 68.7 | 65.2 |
| 29 languages | 72.4 | 67.2 | 56.6 | 60.2 | 63.0 | 54.9 | 55.3 | 59.7 | 70.7 | 90.3 | 67.3 | 65.2 |
| | | | | | | R@100 | | | | | | |
| 11 languages | 83.4 | 89.2 | 55.2 | 81.5 | 83.6 | 75.6 | 72.4 | 75.7 | 88.9 | 70.0 | 91.9 | 78.9 |
| 29 languages | 82.0 | 89.6 | 48.8 | 79.6 | 81.4 | 72.8 | 66.2 | 68.5 | 88.7 | 80.8 | 90.3 | 77.2 |
| *+ MS MARCO* | | | | | | | | | | | | |
| 11 languages | 89.6 | 91.9 | 78.7 | 89.0 | 91.2 | 83.4 | 80.2 | 86.0 | 92.6 | 95.9 | 92.8 | 88.3 |
| 29 languages | 88.7 | 91.4 | 77.2 | 88.1 | 89.8 | 81.7 | 78.2 | 83.8 | 91.4 | 96.6 | 90.5 | 87.0 |
| *+ Mr. TyDi* | | | | | | | | | | | | |
| 11 languages | 94.2 | 98.2 | 93.3 | 93.6 | 94.7 | 89.1 | 87.1 | 92.3 | 94.5 | 98.9 | 96.9 | 93.9 |
| 29 languages | 94.0 | 98.6 | 92.2 | 92.7 | 94.5 | 88.8 | 88.9 | 92.4 | 93.7 | 98.9 | 95.2 | 93.6 |

