# OpenReview forum: "Unsupervised Dense Information Retrieval with Contrastive Learning"
_TMLR — Accepted by TMLR_

### Review · Reviewer_dx1v · 2022-06-14

**Summary Of Contributions:**

* The paper shows that unsupervised (or self-supervised) neural bi-encoder retrievers can do better than previously thought and can be on par with BM25 as measured on the BEIR benchmarks, especially on Recall@100

* A new method for unsupervised contrastive learning of a retriever, previously proposed in the vision community, is applied to text retrieval, and shown to outperform contrastive learning variants used for text retriever training before (e.g. ICT)

* Departures from prior work in terms of training set, batch sizes, and definition of positive/negative examples are ablated and show that MoCo enables the use of larger batch sizes while not hurting performance, which together with cropping to construct examples and use of a different pre-training dataset lead to improvements across many domains

* The paper also shows that the value of unsupervised pre-training can be higher than the value of training on out-of-domain supervised data

* New state-of-the art results on a subset of the BEIR benchmark are achieved, by further finetuning on MSMARCO and re-ranking with a cross-encoder. The comparison is a fair comparison against prior work also using such labeled data and re-rankers.

* Experiments on multilingual retrieval and cross-lingual retrieval are also very strong. Unsupervised contrastive pre-training of the retriever is shown to bring significant value for multilingual retrieval

**Broader Impact Concerns:**

No concern. The work advances the state of the art toward multilingual information access.

**Requested Changes:**

## More substantial

* I would suggest to emphasize the lessons from the ablations more and refer to the ablations section from earlier parts of the paper, because the ablations are the main learnings regarding the impact of differences from prior work. Also when claiming improvements over SOTA, mention the main factors behind those improvements.

* when comparing data augmentation methods in Table 7, I suggest to evaluate ICT with in-batch negatives as well. The authors mention that random cropping is better for MoCo which assumes similar distributions of queries and documents, but we are not sure whether the improved performance of cropping is due solely to the interaction with MoCo

* Gao & Callan 2021  is referred to as concurrent work, but I don't think a difference of 6 months would count as concurrent; please describe differences and learnings compared to Gao & Callan

* P.3 “Instead, we propose to use a bi-encoder architecture, where documents and queries are encoded independently” . This statement seems to imply bi-encoders are newly presented in this work, rephrase to something like “We follow (cite) to use a bi-encoder instead ..”

* Eq (1) on p.4 – the denominator should include the positive example but the description indicates it only sums over a pool of negative documents.

* Mention the pre-training dataset briefly in the main part of the text, e.g. in 4.1 where I currently see only evaluation datasets.

## Less substantial (typos)

Some example typos are below

* P. 1 “billions of element” → billions of elements
* P.2 “uses a given sentence as query and predict the context surrounding it” → “uses a given sentence as a query and predicts the context surrounding it”
* P. 3 “Indices weigh 100GB”
* P.5 “We use these perturbation” → We use these perturbations
* P.12 “neural networks .. exhibits good ..” → “neural networks .. exhibit good ..”






**Strengths And Weaknesses:**

## Strengths
* The work achieves very strong results and advances the state of the art on an important task: being able to to perform high-quality retrieval across domains and languages
* The experiments and approach are described in sufficient detail, although sometimes the information takes a while a to find
* There are good ablations to tease apart the importance of the points of departure wrt prior work (with a small suggestion), and strong baselines from prior work are selected

## Weaknesses

* There is an emphasis on comparing to the state of the art and demonstrating strong performance and it takes effort to find the place where differences to prior work and main factors for the performance differences are described

* The impact of example formulation (in comparison to ICT) is not fully ablated -- I would suggest an additional experiment on that

* When other methods are compared to in figures and Tables, the differences to the proposed method are not articulated

* There are a number of typos which I will list in the requested changes section, some technical imprecision, and imprecision in the claims

---

> ### Author Response · Authors · 2022-06-28
> **Reply to reviewer dx1v**
>
> We would like to thank the reviewer for his comments.
>
> >I would suggest to emphasize the lessons from the ablations more and refer to the ablations section from earlier parts of the paper, because the ablations are the main learnings regarding the impact of differences from prior work. Also when claiming improvements over SOTA, mention the main factors behind those improvements.
>
> The improvement in performance over previous work mainly come from three factors analyzed in the ablation section:
>
> - Using MoCo to handle negatives, which allows to scale to a large number of negatives.
> - The sampling procedure to generate pairs of (query, key).
> - Using data from both CC-net and Wikipedia for training.
>
> We will update the paper to highlight the origin of the gains.
>
>
> > when comparing data augmentation methods in Table 7, I suggest to evaluate ICT with in-batch negatives as well. The authors mention that random cropping is better for MoCo which assumes similar distributions of queries and documents, but we are not sure whether the improved performance of cropping is due solely to the interaction with MoCo
>
> We observed that in-batch ICT is slightly worse than ICT with MoCo. Results are reported in the table below:
> | Model | NFCorpus | NQ | ArguAna | Quora | DBPedia | SciDocs | FEVER | Overall |
> | -----------  | :---: | :---: |:---: |:---: |:---: |:---: |:---: |:---: |
> | MoCo ICT  | 23.2 | 19.4 | 31.6 | 27.6 | 21.3 | 10.6 | 55.6 | 25.9 |
> | In-batch ICT | 20.3 | 13.3 | 23.8 | 69.0 | 20.1 | 9.0 | 38.7 | 24.1 |
>
>
> > Gao & Callan 2021 is referred to as concurrent work, but I don't think a difference of 6 months would count as concurrent; please describe differences and learnings compared to Gao & Callan
>
> The work by Gao & Callan focuses mainly on the advantages of contrastive learning as an in-domain pre-training before fine-tuning. We make relatively similar observations regarding the gains obtained from contrastive learning as a pre-training before fine-tuning. Technically our work differs by the use of MoCo to handle negatives, which allows us to scale to a large number of negatives. This leads to interesting unsupervised performance, on par with BM25 for the Recall@100 on question answering datasets for instance. We also trained a multilingual model with the same contrastive learning approach, obtaining state-of-the-art results after fine-tuning.
> Additionally we would like to point out that our submission is a resubmission that has been public since early October of last year.
>
> > P.3 “Instead, we propose to use a bi-encoder architecture, where documents and queries are encoded independently” . This statement seems to imply bi-encoders are newly presented in this work, rephrase to something like “We follow (cite) to use a bi-encoder instead ..”
>
> We will update the paper to make it more clear that we did not make this contribution.
>
> > Eq (1) on p.4 – the denominator should include the positive example but the description indicates it only sums over a pool of negative documents.
>
> Thank you for catching this! We will update the paper.
>
> > Mention the pre-training dataset briefly in the main part of the text, e.g. in 4.1 where I currently see only evaluation datasets.
>
> We will update the paper to mention the datasets used during pre-training.

---

### Review · Reviewer_CTNi · 2022-06-15

**Summary Of Contributions:**

* The paper presents a new, high-performing dense passage retriever.
* The proposed method is relatively simple, built upon previously introduced techniques for self-supervised learning. From what I understand, two ingredients that make this model work better than the previous iteration of unsupervised retriever -- (1) using MoCo (Momentum Contrast) training to enable large negative batch efficiently and (2) using independent cropping instead inverse cloze task
* The model is evaluated extensively -- on the English QA benchmark, IR benchmark (BEIR), and multilingual QA retrieval benchmark dataset, and shows impressive performance across the board.
* The code/model will be publicly available, and I anticipate many people will use this for future work.


**Requested Changes:**

The experimental setup in this paper is not very clear overall -- few things that need to be clarified:
* The setting of the training data construction setting was not super clear to me. In the end, did you use Crop + delete? Did you also use ICT? What was the final positive training set look like, and how they are sampled?
* I don't understand “+CE” parts on Table 2  — how do you train the cross-encoder here?
---
* I'm not very convinced that crop + delete is better because it provides symmetric views -- I think this could potentially be because it encourages more lexical match. I wonder if increasing the proportion of positive training examples where the query sentence is kept (currently set to 10%) can improve the performance of ICT. As this is one of the main technical differences with prior work, a bit more rigorous study would be helpful.
--
* As mentioned in "weakness", adding a discussion on the computational aspect would be great -- although I assume at inference time the computational costs would be similar to other dense retrieval models.
* Is inverse cloze task / masked salient spans baselines (in table 1) comparable in terms of the amount of training data used?
--
A few missing related work:
* Learning to Retrieve Passages without Supervision: https://arxiv.org/pdf/2112.07708.pdf  --> this is particularly relevant. it's very recent work (which, actually cites results from this paper), but the idea of unsupervised retrieval is very related and the paper will benefit from discussing it together. Overall, I feel this paper didn't really do a great job describing baseline/comparison systems clearly... For example, salient span masking comparison model was introduced a bit out of the blue when discussing baseline/results, maybe mention it a bit when introducing positive training example generation would be helpful.
* Salient Phrase Aware Dense Retrieval: Can a Dense Retriever Imitate a Sparse One? https://arxiv.org/pdf/2110.06918.pdf


More minor comments...
* The supervised results in Table 1 should be updated with more recent numbers, for example.. https://arxiv.org/pdf/2104.08253.pdf
* Why not use nDCG metric for multilingual studies?
* XORQA-Retrieval would be a good benchmark dataset to evaluate cross-lingual retrieval.
* The writing of the introduction can be improved — it’s not super clear how ICT and your approach differ from reading the introduction.
* The second paragraph in the introduction “As training neural networks requires large amount of data” —> no longer true for various NLP tasks which can exploit pre-trained language models.
* I feel abstract is a bit overclaiming — on average it still underperforms BM25.
* I would prefer to have an ablation study earlier than later, probably after when you introduce various design choices.
* On page 5, might be good to provide concrete numbers of necessary batch sizes —  “extremely large batch sizes to work well”.

**Strengths And Weaknesses:**

* Strength:
- Impressive empirical results on important tasks.
- Simple, well-motivated method that works well.

* Weakness:
- There are some issues with clarity in writing, especially with regard to experimental set-ups, see detailed comments in "requested changes" section.
- Comparison with related work/baselines can be much clearer. The work is evaluated in *many* different settings, which is definitely the strength of the paper, but the comparison is often not very clear.. (see requested changes section again).
- Lack of study on computational efficiency: it’d be very helpful to add a discussion on efficiency — both in terms of speed (both training and inference time) and memory requirements. One of the major bottlenecks of building/using retrieval systems is their large memory footprint.

---

> ### Author Response · Authors · 2022-06-29
> **Reply to reviewer CTNi (1/3)**
>
> We would like to thank the reviewer for his comments.
>
> > The setting of the training data construction setting was not super clear to me. In the end, did you use Crop + delete? Did you also use ICT? What was the final positive training set look like, and how they are sampled?
>
> For the Contriever model, we use random cropping with 10% of token deletion. Except for the ablations, we don't use ICT. To sample a positive pair (query, key) we start from a list of 256 tokens obtained by applying the BERT BPE tokenizer. Then, we sample the length of the query and the length of the key uniformly between 12 and 128 tokens. Finally, we uniformly sample the query and the key as contiguous segments of tokens according to their respective length among the 256 tokens. Below is an example of (key, query) pair.
>
> Query: *[CLS] lighting designer – billy mawer an early review of a technical rehearsal, published three days prior to the opening of the show to the public was negative – " not quite the next rocky horror ", however subsequent media responses based on the actual performances before paying audiences were all positive. the script and playing time was reduced by 22 minutes prior to opening night and production was tightened in the first week of the season. the show went on to receive positive responses from many sources. these include : alan jones - " there ’ s a rock musical on [SEP]*
>
> Key: *[CLS] prior to opening night and production was tightened in the first week of the season. the show went on to receive positive responses from many sources. these include : alan jones - " there ’ s a rock musical on at the moment called the island of doctor moron … the singing dancing is unbelievable. " ; " the island of doctor moron has come [SEP]*
>
> > I don't understand “+CE” parts on Table 2 — how do you train the cross-encoder here?
>
> We used the cross-encoder from the original BEIR paper, used for BM25+CE baseline and available here:  https://huggingface.co/cross-encoder/ms-marco-MiniLM-L-6-v2. Our goal with this experiment is to illustrate that the strong recall obtained by our retriever translates to state-of-the-art results when used with a cross-encoder reranker. More generally, we argue that to obtain strong nDCG results, the current best way is to use a two stage reranking approach, where the first stage should have the best possible recall.
>
> > I'm not very convinced that crop + delete is better because it provides symmetric views -- I think this could potentially be because it encourages more lexical match. I wonder if increasing the proportion of positive training examples where the query sentence is kept (currently set to 10%) can improve the performance of ICT. As this is one of the main technical differences with prior work, a bit more rigorous study would be helpful.
>
> This experiment was previously conducted in the paper introducing ICT, and we re-used the optimal parameter that they found (Fig. 3 from https://arxiv.org/abs/1906.00300).

---

> > ### Author Response · Authors · 2022-06-29
> > **Reply to reviewer CTNi (2/3)**
> >
> > > As mentioned in "weakness", adding a discussion on the computational aspect would be great -- although I assume at inference time the computational costs would be similar to other dense retrieval models.
> >
> > The majority of the methods we compare to are retrievers using dense bi-encoders based on variants of BERT base. Thus embedding the collection of documents in which retrieval is performed will have the same cost.
> >
> > At inference these methods will all benefit from techniques aiming at reducing the memory or computational cost of nearest neighbour search. Among others, product quantization and dimension reduction have been used to reduce memory requirements, and approximate nearest neighbour search has been used to speed up retrieval.
> >
> > Among the methods we compare to, there are several methods that are not using a standard dense bi-encoders framework:
> > - Colbert uses an enhanced mechanism to compare query and document representations, this will typically require more memory and lead to slower inference.
> > - Splade and sparse methods such as BM25 are generally relatively fast but they are difficult to compare with dense retrievers because they don’t benefit from approximate nearest neighbor search or quantization of the index.
> > - DocT5query uses a T5 encoder-decoder to augment the query.
> >
> > Regarding the cost of the pre-training: the contrastive pre-training is performed for 500k steps with a batch size of 2048 with on average 128 tokens per example. This is approximately the number of tokens used to train BERT. In the case of our contrastive pre-training most of the performance is already obtained with 20% of the pre-training as reported on QA datasets in the table below. Furthermore, the additional cost of pre-training is amortised if the model is fine-tuned on multiple datasets, similarly to the additional cost of pre-trained models such as BERT.
> >
> > > Is inverse cloze task / masked salient spans baselines (in table 1) comparable in terms of the amount of training data used?
> >
> > Details of the ICT/MSS baselines are reported in Appendix A.2 of the corresponding paper. These baselines are trained for 100k steps with a batch size of 4096 and a maximum sentence length of 256 tokens. This leads to a rough upper bound of 210B tokens, but doesn't take into account the fact that on average the sentence length is shorter than 256 tokens. On our side we report our final results with 500k steps with a batch size of 2048 and on average  maximum sentence length of 140 tokens per sample, roughly corresponding to 143B tokens.
> >
> > To further illustrate the benefits of our approach, we report results by training our model for 100k steps only (instead of 500k), leading to a performance slightly lower than our full training. Notably, it still outperforms the ICT/MSS baselines, while having a strictly lower computational cost. We also report results obtained using only Wikipedia data similarly to the ICT/MSS baselines.
> >
> > | Model | NQ R@5 | NQ R@20 | NQ R@100 | TQA R@5 | TQA R@20 | TQA R@100 |
> > | -----------  | :---: | :---: |:---: |:---: |:---: |:---: |
> > | Inverse Cloze Task (Sachan et al.) | 32.3 | 50.9 | 66.8 | 40.2 | 57.5 | 73.6 |
> > | Masked salient spans (Sachan et al.) | 41.7 | 59.8 | 74.9 | 53.3 | 68.2 | 79.4|
> > | BM25 (Ma et al.) | - | 62.9 | 78.3 | - | 76.4 | 83.2 |
> > | Contriever | 47.8 | 67.8 | 82.1 | 59.4 | 74.2 |83.2 |
> > | 100k training steps on CC-net + Wikipedia | 46.0 | 67.1 | 81.7 | 60.0 | 74.2 |83.1 |
> > | 100k training steps on Wikipedia | 42.7 | 63.4 | 77.7 | 55.9 | 70.9  | 81.8 |

---

> > > ### Author Response · Authors · 2022-06-29
> > > **Reply to reviewer CTNi (3/3)**
> > >
> > > >A few missing related work:
> > > Learning to Retrieve Passages without Supervision: https://arxiv.org/pdf/2112.07708.pdf --> this is particularly relevant. it's very recent work (which, actually cites results from this paper), but the idea of unsupervised retrieval is very related and the paper will benefit from discussing it together. Overall, I feel this paper didn't really do a great job describing baseline/comparison systems clearly... For example, salient span masking comparison model was introduced a bit out of the blue when discussing baseline/results, maybe mention it a bit when introducing positive training example generation would be helpful.
> > > Salient Phrase Aware Dense Retrieval: Can a Dense Retriever Imitate a Sparse One? https://arxiv.org/pdf/2110.06918.pdf
> > >
> > > We will add references to the missing related work. We would like to point out that our submission is a resubmission that has been public since early October of last year, similarly to the first paper mentioned, the second one also cites our work and uses our pre-trained Contriever model.
> > >
> > > > The supervised results in Table 1 should be updated with more recent numbers, for example.. https://arxiv.org/pdf/2104.08253.pdf
> > >
> > > We will update our paper to add state-of-the-art supervised results on NQ and TQA.
> > >
> > > > Why not use nDCG metric for multilingual studies?
> > >
> > > For Mr. TyDi we follow the original for the use of MRR@100 and R@100. For the cross-lingual setup, the retrieval dataset is derived from a question answering dataset, in this area Recalls are the prevalent metrics. This likely comes from the fact that generally a complete pipeline for question answering takes as input retrieved passages to generate the answer ignoring the ranking of the retriever.
> > >
> > > > XORQA-Retrieval would be a good benchmark dataset to evaluate cross-lingual retrieval.
> > >
> > > Thank you for the suggestion. We mainly used a derivative of MKQA because the set of included languages is larger and differs from Mr. TyDi. The fact that MKQA provides the same query translated in multiple languages also offers a way to compare the capacity of cross-lingual retrieval across different languages.
> > >
> > > > The writing of the introduction can be improved — it’s not super clear how ICT and your approach differ from reading the introduction.
> > >
> > > We will update the introduction to make it clearer.
> > >
> > > > The second paragraph in the introduction “As training neural networks requires large amount of data” —> no longer true for various NLP tasks which can exploit pre-trained language models.
> > >
> > > We will modify the sentence to make it specific to the context of information retrieval.
> > >
> > > > I feel abstract is a bit overclaiming — on average it still underperforms BM25.
> > >
> > > We’ll emphasize the fact that unsupervised performances are competitive with BM25 specifically for the Recall@100 in the abstract.
> > > The fact that the unsupervised model underperforms BM25 is true for metrics measuring the quality of the very first retrieved documents. For the Recall@100 we don’t believe that it’s the case. On the BEIR benchmark the difference on average comes from two datasets with specific features as explained in page 8. While on question answering datasets, both on English and multilingual datasets, our models tend to outperform BM25 in average.
> > >
> > > > On page 5, might be good to provide concrete numbers of necessary batch sizes — “extremely large batch sizes to work well”.
> > >
> > > We will update the paper with the relevant information.

---

### Review · Reviewer_aLmD · 2022-06-15

**Summary Of Contributions:**

Recent studies on document retrieval have shown that dense pre-trained retrievers on a large retrieval dataset tend to be outperformed by classical frequency-based methods when tested on new domains and such datasets are hard to collect for different languages. This paper proposes to train dense retrievers in an unsupervised way (without relying on labeled data) using contrastive learning on an auxiliary task that approximates retrieval. Evaluation on few-shot settings shows competitive performance compared to pretraining on large labeled datasets during pretraining. It also appears to be the first example of reaching competitive performance to classical methods when pre-trained on large datasets.

**Broader Impact Concerns:**

The work relies on big pre-trained models for which there are ethical concerns regarding discrimination, exclusion, misinformation, malicious uses, and so on. Previously they have been used for document retrieval but it would be good to add a broader impact statement to discuss such concerns in this area.

**Requested Changes:**

The description should emphasize the novel aspects and make clear what parts are new compared to key most related prior work such as Lee et al. (2019) beyond the empirical results.

The method section has some phrasing that suggests existing ideas are proposed for the first time e.g. using the independent encoding of the query and document which is a bit misleading (e.g. sBERT). For contrastive learning, I'd suggest citing the ICT paper which used a similar objective, and then pointing out the differences.

For the multilingual experiment, the authors should clarify whether the mBERT baseline is pre-trained on the same data as the proposed method. If not, I would suggest adding this comparison for completeness.

In experiments that require finetuning, it would be interesting to report what is the parameter count for each method where applicable to get a better understanding of the differences.



**Strengths And Weaknesses:**

**Strengths**

It proposes and develops an interesting unsupervised method for dense retrieval based on contrastive learning that is inspired by data augmentation methods in vision. The novelty with respect to prior work by Lee et al. (2019) is the way positive examples are built from a document, namely by sampling two different text spans allowing for overlap as opposed to contiguous, mutually exclusive spans.

The empirical investigation is thorough and covers a range of experiments ranging from few-shot retrieval, zero-shot and multilingual retrieval with competitive results in all cases. This provides solid evidence for the effectiveness of the approach and it also provides ablation experiments that justify the design choices.  The comparison includes previous data augmentation approaches for similar tasks based on inverse cloze tasks.

Writing is clear and the general positioning with respect to prior work is quite good. It is important to emphasize classical baselines for these tasks and improving over them is important and should be of interest to the relevant retrieval community.

**Weaknesses**

Conceptually the proposed idea is similar to existing techniques and it could be viewed as incremental. For instance, the bi-encoder architecture is pretty common, with contrastive loss, and negative handling.

Improvements in multilingual retrieval appear to be based on pretraining on 29 languages which can in principle be applied to mBERT. Even though the gap between them is big, it is not clear if the improvement is due to the additional data or the superiority of the learning contrastive objective.

---

> ### Author Response · Authors · 2022-06-28
> **Reply to reviewer aLmD**
>
>
> We would like to thank the reviewer for his feedback.
>
> > The description should emphasize the novel aspects and make clear what parts are new compared to key most related prior work such as Lee et al. (2019) beyond the empirical results.
>
> The improvement in performance over ICT mainly comes from three factors analysed in the ablation section:
>  - Using MoCo to handle negatives, this allows to scale to a large number of negatives.
>  - The sampling procedure to generate pairs of (query, key).
>  - Using data from both CC-net and Wikipedia for training.
>
> We will update the paper to highlight the origin of the gains.
>
> > The method section has some phrasing that suggests existing ideas are proposed for the first time e.g. using the independent encoding of the query and document which is a bit misleading (e.g. sBERT). For contrastive learning, I'd suggest citing the ICT paper which used a similar objective, and then pointing out the differences.
>
> We will rephrase the introduction of the bi-encoder, and make it clear that ICT also uses the infoNCE loss standardly used for contrastive learning.
>
> > For the multilingual experiment, the authors should clarify whether the mBERT baseline is pre-trained on the same data as the proposed method. If not, I would suggest adding this comparison for completeness.
>
> The mBERT baseline is pre-trained on Wikipedia data only, while our model is obtained by further training mBERT on Wikipedia and the CC100 data from XLM-R. We report in the following table a comparison to the XLM-R base model, fine-tuned on MS MARCO, on the Mr.TyDi benchmark. One can note that our model outperforms both the mBERT and XLM-R baselines, which obtain similar results. The strong performance of our model can thus be attributed to the training objective, rather than the data.
>
> | Model | ar | bn | en | fi | id | ja | ko | ru | sw | te | th | avg |
> | -----------  | ----------- | -----------  | ----------- | -----------  | ----------- | -----------  | ----------- | -----------  | ----------- | -----------  | ----------- | -----------  |
> | MRR@100 |
> | mBERT + MSMARCO | 33.9 | 36.2 | 23.9 | 30.0 | 35.2 | 26.3 | 28.6 | 30.3 | 36.9 | 38.9 | 18.7 | 30.8 |
> | XLM-R + MSMARCO  | 25.0 | 35.2 | 23.9 | 26.3 | 28.4 | 20.1 | 26.8 | 25.7 | 29.7 | 26.2 | 29.5 | 27.0 |
> | mContriever + MSMARCO  |  43.4 | 42.3 | 27.1 | 35.1 | 42.6 | 32.4 | 34.2 | 36.1 | 51.2 | 37.4 | 40.2 | 38.4 |
> | R@100 |
> | mBERT + MSMARCO | 81.1 | 88.7 | 78.3 | 74.2 | 81.0 | 76.1 | 66.7 | 77.6 | 74.1 | 89.5 | 57.8 | 76.8 |
> | XLM-R + MSMARCO | 80.6 |  87.4 | 75.8 | 80.8 | 87.9 | 71.0 | 69.6 | 76.2 | 72.7 | 89.9 | 90.4 | 80.2 |
> | mContriever + MSMARCO  | 88.7 | 91.4 | 77.2 | 88.1 | 89.8 | 81.7 | 78.2 | 83.8 | 91.4 | 96.6 | 90.5 | 87.0 |
>
>
> The results reported here for mBERT + MSMARCO differ from the results reported in the paper because the temperature used to fine-tune the model is different from the temperature previously used. Fine-tuning XLM-R with the temperature previously used in the paper led to poor performance. A similar temperature temperature tends to work well for fine-tuning XLM-R and mBERT on MSMARCO before testing on Mr. TyDi. The pre-trained model, mContriever, is not sensitive to the temperature used during fine-tuning. We originally used 0.05 similarly to the temperature used for the English model and didn’t try any other value for this hyperparameter. We will update the paper to include this change.
>
> > In experiments that require finetuning, it would be interesting to report what is the parameter count for each method where applicable to get a better understanding of the differences.
>
> All methods using neural networks are based on the BERT base model, aside of ANCE which uses RoBERTa-base, and docT5query which uses an additional T5 model. Thus the parameter count of the underlying model is roughly the same for all fine-tuned models.

---

### Decision · Action_Editors · 2022-07-14

**Recommendation:** Accept with minor revision

**Comment:**

Reviewers all agreed that this submission should be accepted. Please make the updates you discussed with the reviewers, taking particular heed to clarify novelty w.r.t. existing papers and update the multilingual experiments.